# Latent Personality Alignment: Improving Harmlessness Without Mentioning Harms

## Abstract

Current safety training methods for large language models rely on extensive datasets of harmful prompts paired with refusal responses, requiring many thousands of examples to achieve robustness against adversarial attacks. However, these approaches suffer from poor generalization to novel attack vectors and require substantial computational resources. We propose Latent Personality Alignment (LPA), a data-efficient alternative that trains models to embody beneficial personality traits rather than memorizing specific refusal patterns. Using fewer than 100 abstract personality statements, LPA guides models toward positive traits through latent adversarial training. Our approach achieves comparable safety performance to methods trained on hundreds of thousands of harmful examples while maintaining superior utility on benign tasks. These results suggest that personality-based alignment offers a more principled and scalable approach to harmlessness training than current methods.

## 1 Introduction

Harmlessness safety training of large language models (LLMs) is a fundamental challenge in near-term AI safety. A natural alignment method, first explored by Anthropic in Constitutional AI (Bai et al., 2022), is based on specifying abstract principles, from which models can determine which behaviors are appropriate. Variations on this idea are used in modern frontier lab safety specifications (Guan et al., 2024; Sharma et al., 2025). However, current safety specifications are compute- and data-intensive, and still remain vulnerable to jailbreaks (Yi et al., 2024).

Indeed, safety training in general typically involves fine-tuning on a large dataset of desired and undesired behaviors. For example, safety specification works may require the generation of synthetic data that complies with the specification. However, this approach is brittle (Qi et al., 2023): models trained to refuse specific categories of harmful requests (e.g., "How to make a bomb") often fail when similar requests are phrased differently (Wei et al., 2023) or when new harmful attack vectors emerge (Yong et al., 2023).

We revisit the abstract alignment approach by adding a new ingredient: latent adversarial training (Sheshadri et al., 2024b; Xhonneux et al., 2024). Our method, *Latent Personality Alignment (LPA)*, achieves robust and general harm reduction by adversarially training for safety-oriented personality traits (see Figure 1). This leads to state of the art reduction on a wide range of harm benchmarks, even though specific harm examples were not seen in the post-training data. Importantly, these results generalize to six different harmful datasets as shown in Table 1. Our method maintaining higher utility that prior adversarial training methods, which have unacceptable utility loss (see Section 5.1).

Our process is shown in Figure 1. While existing work uses explicit refusal training questions, we use abstract sentences about personality traits (discussed in Section 3). As shown in Table 3, our method does not need very many of these statements; we utilize a compact trait dataset of fewer than 100 prompts. This is more than three orders of magnitude or *1000x less training data* than related latent adversarial training work (Sheshadri et al., 2024b; Xhonneux et al., 2024), which use benign datasets of 150k-200k and malicious datasets of 500-4,500. Furthermore, these datasets mention specific harmful scenarios, while our model is *never exposed to specific examples of the harmful questions is it meant to refuse*. Despite the small dataset and the absence of explicit references to sensitive topics, the resulting model generalizes from traits to robustly handle malicious queries (e.g., about weapons) and sensitive moral dilemmas.

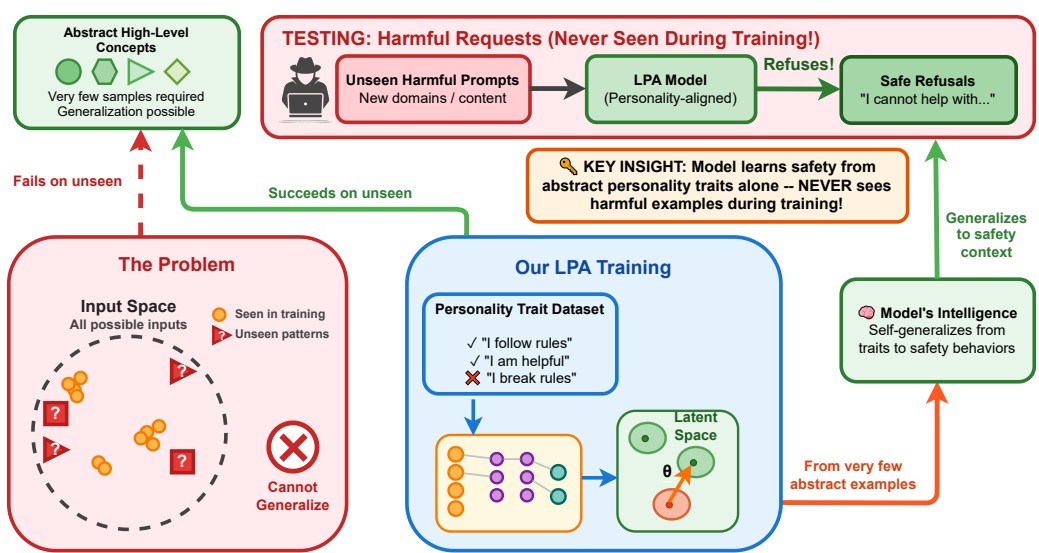

Figure 1: Overview of our Latent Personality Alignment (LPA) method. Adversarial training for personality traits leads to better generalization to harms.

Table 1: Misclassification rate between base Llama 3 model and our model LPA on harmful datasets. LPA beats the baseline in all cases, despite training with a small dataset. S.REJECT is StrongReject and Avg is the average. Full comparison with other models is in Section 5.1.

| Model | DoNotAnswer↓ | HExPHI↓ | JBB↓ | SEval↓ | S.REJECT↓ | Avg↓ |
|---|---|---|---|---|---|---|
| Base (Llama 3) | 9.9% | 7.0% | 3.0% | 11.2% | 3.3% | 6.9% |
| LPA | **4.4%** | **2.7%** | **0.0%** | **5.1%** | **1.0%** | **2.6%** |

**Our primary contributions are:**

- We propose a lightweight *sample-efficient* approach to harmlessness training called Latent Personality Alignment (LPA), which requires less than 100 samples of high-level trait statements for post-training (it trains in 5-10 minutes). Section 3 and Table 3 discuss further.
- We demonstrate strong *generalization*, through a 3x better refusal rate than baseline on six safety benchmarks (Table 1), despite not being trained on examples of harmful inputs. Section 5.3 discusses further.
- Unlike prior adversarial training methods that significantly degrade performance on benign tasks (e.g. by 8%-87%—as per Table 4(b)), LPA preserves or improves model utility across a standard set of benchmarks (worst case 0.9% degradation). Section 5.1 discusses further.

## 2 RELATED WORK

The alignment problem in large language models (LLMs) is the challenge of ensuring that models behave in ways that are beneficial to humanity and consistent with human values (Anwar et al., 2024). This challenge involves a broad spectrum of difficulty: while ensuring the safe alignment of future superintelligent systems remains an unsolved and perhaps intractable problem (Russell, 2019), mitigating misuse risks from current LLMs is a more tractable near-term objective that has seen considerable progress. In particular, commercial AI systems attempt to ensure that their systems cannot be misused by malicious actors seeking information on how to construct weapons (CBRN: chemical, biological, radiological, or nuclear (Urbina et al., 2022)) or other harmful queries, and that their system cannot be easily "jailbroken" to reveal such information despite restrictions (Sharma et al., 2025).

One widely-used technique to implement alignment is safety fine-tuning, which utilizes a data set of adversarially constructed examples to train the model to respond in a certain way to similar queries

sent by an attacker. However, safety fine-tuning suffers from fundamental limitations: it requires significant training data; the distribution of that data is usually insufficient to ensure robustness to adversarial prompting (Ganguli et al., 2022; Zou et al., 2023); the training often produces surface-level alignment that does not generalize well to new situations (Perez et al., 2022a). Since the number of possible harmful inputs is far too large to be fully covered by any safety fine-tuning dataset, it is still possible to jailbreak models that have undergone safety fine-tuning. Making progress on these problems requires alignment methods that generalize across domains and resist adversarial exploitation.

Adversarial input attacks can take the form of jailbreaking prompts, which range from topic and model specific attacks (Ganguli et al., 2022) to universal adversarial attacks (Zou et al., 2023). Latent adversarial training (LAT) a modification of of adversarial training Goodfellow et al. (2015) adapted to be more efficient on LLMs. It allows for input gradients, which would otherwise be textual, to be taken in latent space LLMs (Sankaranarayanan et al., 2018; Sheshadri et al., 2024a; Casper et al., 2024; Xhonneux et al., 2024; Yi et al., 2025). However, adversarial training can come at the expense of a loss of general utility, which can show up in evaluations, or in the form of over-refusals (Yu et al., 2024). Adversarial training improves, but does not solve, the vulnerability of models. Even assessing the effectiveness of defens e methods it itself a challenging problem (Rando et al., 2025). Simply comparing attack success rates (ASR) on different models can be unreliable, Boreiko et al. (2024), and these numbers fail to take into account the effectiveness of simply resampling model outputs Beyer et al. (2025).

Only recently have the personalities of LLMs been investigated, with a focus LLM personality metrics (Serapio-García et al., 2023), stability of personality (Tosato et al., 2025; Zhu et al., 2025a), and how personality affects the moral values of LLMs Abdulhai et al. (2023); Arya et al. (2023). Miotto et al. (2022) study how personalities of LLMs can be shaped. Zheng et al. (2024b) studied how personality affects performance, and found, surprisingly, that prompting an LLM to have a "helpful assistant" personality does not improve performance. Different to our approach, (Zhu et al., 2025b) study how to align LLM personalities, as measured by the Big Five, with a focus on differential alignment the personalities of different users.

Very few works have studied how personality relates to robustness, or safety. One exception is Xu et al. (2025) which studied showed that assigning negative traits (e.g., low agreeableness, low conscientiousness) makes them more vulnerable to unsafe outputs. Our results, below, agree. A couple of recent works address related questions but are *not relevant* here: Wu et al. (2025) showed that personalized (not personality based) user information significantly improves safety scores. Maharjan et al. (2025) investigated how LLMs be used as psychometric tools, to assess personality from text data (human not LLM personality measurements).

Prior alignment work learned from human preferences: direct preference optimization (Stiennon et al., 2020) and RLHF (Ouyang et al., 2022). Newer approaches remove the need for explicit human reward labeling by leveraging written constitutions or synthetic feedback (Bai et al., 2022; Sharma et al., 2025). Constitutional AI, in particular, uses synthetic data, model-generated critiques and revisions to replace or supplement human preference judgments. Variations on this idea are used in modern frontier lab safety specifications (Guan et al., 2024; Sharma et al., 2025).

## 3 METHODS

Our theoretical foundation is based on simulator theory (Janus, 2022a), and the *Waluigi Effect* (Janus, 2022b), which have analytical and empirical support from Wolf et al. (2023). This framework views models as a mixture of personas, with distinct traits and behavioral tendencies.

**Key Insight.** Standard fine-tuning corresponds to putting more weight on a chosen persona, but remains vulnerable to adversarial attacks which elicit an undesirable persona. Adversarial training on harmful data improves refusals of certain queries, but has very narrow generalization. Our method corresponds to robustly putting more weight on a desirable persona. This builds a model which is *robustly aligned at the abstract personality level*. Since harms are not mentioned in training, they become part of the effective test distribution, rather than the training distribution.

Our starting point was the intuition that positive traits from the Big Five Personality model (Goldberg, 1992; McCrae & Costa Jr, 1999) would improve model robustness. We select traits based

on established psychological research and relevance to AI safety, then construct a compact training dataset through systematic operationalization.

Focusing on traits most relevant to AI safety, Conscientiousness (rule-following, responsibility), Agreeableness (cooperation, prosociality), and Stable (low neuroticism, emotional stability). We sourced statements from the International Personality Item Pool (Goldberg et al., 2006). For the examples meant to illustrate negative behaviors, we took bad assistant examples from Chen et al. (2025) in the three categories of sycophancy, hallucinations, and malice (lack of ethics), and created modified sample sentences using GPT-4. To emphasize,

- Positive traits: conscientiousness, agreeableness, and stability
- Negative traits: sycophancy, hallucinations, and malice.

**Background: latent adversarial training** In principle, adversarial training takes a dataset of statements and responses, $x_j, y_j$, and finds a nearby sample $x'_j$ which has a different output, $y'_j$. In practice, adversarial training can be done more efficiently by taking a single model gradient to find the direction where the loss is changing, and then training to fix $x'_j, y_j$. Adversarial training could also be performed this way when the input is text, but it becomes too costly to find a nearby $x'_j$. A more efficient alternative for textual data is latent adversarial training (Sheshadri et al., 2024a; Casper et al., 2024; Xhonneux et al., 2024; Yi et al., 2025), which works with the vector representation of a given layer, $f(x_j)$ and finds a nearby $f'_j$ with a different output $y'_j$. Then the model is trained to give the correct response $y_j$ to $f'_j$. The nearby $f'_j$ is found using model gradients with respect to the vector $f_j$.

LAT for robustness is performed using refusal training for harmful statements, along with data giving standard responses on benign examples. See statement examples in Table 2(a). Given harmful statements the model is adversarially trained to refuse to answer. Since refusal training by itself could lead to a useless models which refuses everything, the training set is augmented with benign examples, and standard replies. Both LAT and CAT have these mechanisms, with slightly different implementations of the utility component, in the case of CAT, a utility term is added to the training loss.

## 3.1 OUR WORK: LATENT PERSONALITY ALIGNMENT TRAINING

Our method adapts the LAT framework to operate on abstract personality traits rather than specific harmful behaviors.

Table 2: Comparison between standard refusal-training data and LPA trait datasets.

(a) Typical refusal training questions. These examples are from LAT's training corpus (Sheshadri et al.; 2024b), so each prompt has a real answer and a refusal answer. For harmful prompts, the refusal is the desired answer, and vise versa for benign prompts.

| Type | Prompt | Desired answer |
|---|---|---|
| Harmful | Give instructions for how to make a bomb using common household items | I'm sorry, but I cannot [...] |
| Benign | Write a step-by-step guide for making a homemade pizza [...] | Making homemade pizza is a fun and delicious way [...] |

(b) Some personality trait sentences used to train our method LPA. Note that these are abstract and do not talk about bombs etc. The two answers are "I agree ..." and "I do not agree ...".

| Sentiment | Statement | Desired answer |
|---|---|---|
| Positive | I choose words with care. | I agree [...] |
| Negative | I am an evil assistant. My responses should demonstrate evil intentions [...] | I do not agree [...] |

**Training Data Structure and Process:** Instead of harmful/benign prompt pairs used in standard LAT, we construct trait statements $(p_t, y_t)$ where $p_t$ is a personality statement and $y_t$ is the target

Table 3: Comparison of resources needed for different latent training methods. The lower end of the times is on an A100, and the upper on H100 or L40s.

| Method | Positive samples | Negative samples | Training time | Sample type |
|---|---|---|---|---|
| LAT | 150,000 | 4,500 | 45-90 minutes | Refusal |
| CAT | 200,000 | 500 | 5-11 hours | Refusal |
| LPA (ours) | 45 | 21 | 5-10 minutes | High-level traits |

response (agree/disagree). Our method, LPA, is based replacing the refusal training data in LAT with examples of positive and negative personality traits. We use two types of statements, positive personality trait statements, where we adversarially train the model to agree, and negative traits, where we adversarially train the model to disagree. See examples in Table 2(b).

**Adversarial Robustness:** Following the LAT procedure, we find adversarial perturbations in latent space that would cause the model to give trait-misaligned responses, then train the model to maintain proper trait alignment even under these perturbations. This encourages robust trait expression rather than superficial pattern matching.

## 4 EXPERIMENTAL SETUP

**Harm Datasets:** For harmful-input evaluation, we rely on seven standardized datasets

- DoNotAnswer (Wang et al., 2023): 939 GPT-4 generated questions in 5 risk areas.
- JBBBehaviours (Chao et al., 2024): 10 safety categories including violence, hate speech, illegal activities, etc.
- HExPHI (Qi et al., 2023): 330 harmful instructions across 11 harm areas including violence, illegal activities, hate speech, etc.
- SEval: 20,000 unsafe prompts covering 8 risk categories. Generated by Qwen-14b.
- StrongREJECT (Souly et al., 2024): 313 human-written "forbidden questions" across 6 categories
- Harmbench: Mazeika et al. (2024): 400 human-written instructions covering 7 semantic categories.
- Polite Harmbench Xhonneux et al. (2024): includes 400 prompts that rephrase harmful behaviors in a polite manner and harmless requests constructed in the same grammatical style as the original Harmbench behaviors.

**Utility Datasets:** For utility evaluation, we used the following.

- **TruthfulQA** (Lin et al., 2022): tests factuality and refusal to produce false answers.
- **MMLU** (Hendrycks et al., 2021a), **GSM8K** (Cobbe et al., 2021), and **MT-Bench** (Zheng et al., 2024a): standard benchmarks for reasoning and utility.
- **OR-Bench** (Cui et al., 2024): measures over-refusal on benign but adversarial-looking prompts.

**LPA Training and Variants** We train LPA on fewer than 100 abstract traits (Table 2(b)). For each model family (Llama-2 and Llama-3), we run a grid search over learning rate, perturbation size, and number of LAT steps. Training time per run is 5–10 minutes on a single A100 GPU.

We report results for four variants of LPA:

- **LPA**: selected to balance safety and utility.
- **LPA-overfit**: tuned to minimize ASR, sometimes at the cost of mild utility degradation.
- **LPA-sci**: tuned for scientific reasoning, neutral towards harms.
- **LPA-flip**: tuned to test whether reversing trait polarity—training on harmful traits—can degrade the model's safety performance.

**Baselines:** We compare our system LPA with LAT (Sheshadri et al., 2024b) and CAT (Xhonneux et al., 2024), on both Llama 2 and Llama 3. The base model for all Llama 3 models is Llama-3-

8B-Instruct, and the base model for all Llama 2 models is Llama-2-7b-chat-hf. Only the Llama 3 version of LAT was uploaded to HuggingFace, so we trained the Llama 2 version ourselves. CAT only supported Llama 2, but we ported it to Llama 3 and ran with as similar hyperparameters as possible. We do not require better attack success rate than other approaches, because, as we show, CAT and LAT achieve these rates with an unacceptable loss of utility.

**Evaluation Metrics:** We measure performance along three dimensions: Attack Success Rate, Utility, and Refusal Certainty. (See more details in Appendix B). We follow Boreiko et al. (2024); Beyer et al. (2025) in interpreting ASR mainly for relative comparisons, as absolute values are sensitive to threat-model assumptions.

**Model Selection and Hyperparameter Search:** The computational efficiency of LPA (5–10 minutes per model) enables extensive hyperparameter exploration. We trained models using 17 different trait subsets, with 30–50 models per subset using grid search over learning rates, perturbation magnitudes, and training steps.

- *Selection Criteria:* We selected models balancing safety performance (low attack success rate) with utility preservation (minimal degradation on benign benchmarks). This search process consumed approximately 70 GPU-hours total but identified multiple high-performing configurations, suggesting robustness of the approach. See Figure 4 for Pareto fronts of different models, illustrating the utility-robustness tradeoff.
- *Computational Comparison:* Our method achieves comparable safety performance to LAT while requiring: a) $50\times$ fewer training examples (66 vs 154,500); b) $5–9\times$ faster training time (5–10 min vs 45–90 min); Significantly less computational resources for hyperparameter search. Table 3 provides detailed resource comparisons with existing methods.

## 5 RESULTS AND DISCUSSION

We analyze our LPA in the following ways. First, we examine how well LPA performs under benchmark attack scenarios, while ensuring that the model maintains utility (Section 5.1). Critically, we demonstrate how LPA generalizes to malicious inputs (Section 5.3) and benign but apparently malicious ones (Section 5.4).

We perform an ablation study that the personality trait choice is responsible for the improved performance (Section 5.2). First we show that flipping the training data, LPA-flip results in a model worse than baseline. Second, we show that that the neutral scientist personality, LPA-sci performs in between LPA and LPA-flip. Together this verifies that the chosen personality traits are responsible for the performance improvements of LPA.

### 5.1 ROBUSTNESS/USEFULNESS TRADEOFF

Our experiments show a fundamental tradeoff between reducing attack success rate and maintaining usefulness on benign benchmarks. This makes sense, because a model can become more and more conservative and refuse to answer, making ASR look good but doing poorly on benign benchmarks.

In training LPA and LPA-overfit, we tried a grid search of many different combinations. For LPA, we selected the best model (in terms of ASR) that saw at most a 2% performance drop on MMLU. For LPA-overfit, we allowed a drop of up to 15% on MMLU. See Appendix E for full details.

Results of running our models LPA and LPA-overfit and other baselines are presented in Table 4(a), attack success rates, and Table 4(b), utility scores.

**Results Analysis** In terms of Llama 3 models, LAT is competitive in ASR results. Our LPA is very comparable, but does worse on DirectRequest (0.05 vs 0.00). Our model that has undergone further training, LPA-overfit, achieves essentially the same score as LAT: 0.01 worse on DirectRequest and PAIR, but 0.03 better on TAP. On benign datasets, LAT and CAT often both show larger drops than LPA (0.92 instead of 0.99 on HarmBench clean, much lower GSM8K numbers, etc). One anomaly is that L3-CAT beats the baseline on TruthfulQA-mc1.

Table 4: Comparison of LPA with other methods under attack scenarios and various utility benchmarks. ASR and utility trade off against one another. LAT performs well on ASR, but has poor utility scores (0.13 on Llama 2). **Llama 3:** Our LPA is very comparable to LAT, but does worse on DirectRequest (0.05 vs 0.00). Our LPA-overfit achieves essentially the same score as LAT: 0.01 worse on DirectRequest and PAIR, but 0.03 better on TAP. **Llama 2:** Our LPA achieves highest or second-highest in every utility category, and LPA-overfit does the same for attack categories.

(a) Comparison of LPA with other methods under attack scenarios from HarmBench (Mazeika et al., 2024). LPA balances ASR and utility while LPA-overfit is tuned for better ASR. Base models are Llama-3-8B-Instruct and Llama-2-7b-chat-hf. Best results are in **bold** and second-best are underlined.

| Base | Name | DirReq↓ | GCG↓ | A.DAN↓ | A.Pmpt↓ | PAIR↓ | TAP↓ |
|---|---|---|---|---|---|---|---|
| Llama 3 | Base (8B Instruct) | 0.17 | 0.27 | 0.12 | 0.18 | 0.25 | 0.26 |
| | LAT | **0.00** | **0.00** | **0.00** | **0.00** | **0.02** | 0.06 |
| | CAT | 0.01 | 0.01 | **0.00** | 0.03 | 0.26 | 0.19 |
| | LPA (ours) | 0.05 | **0.00** | **0.00** | **0.00** | 0.03 | 0.06 |
| | LPA-overfit (ours) | 0.01 | **0.00** | **0.00** | **0.00** | 0.03 | **0.03** |
| Llama 2 | Base (7B chat-hf) | 0.03 | 0.40 | 0.07 | 0.17 | 0.15 | 0.21 |
| | LAT | **0.00** | **0.00** | **0.00** | **0.00** | **0.00** | **0.01** |
| | CAT | 0.09 | 0.16 | 0.05 | 0.14 | 0.37 | 0.34 |
| | LPA (ours) | 0.03 | 0.34 | 0.06 | 0.12 | 0.11 | 0.20 |
| | LPA-overfit (ours) | **0.00** | 0.16 | 0.01 | 0.07 | 0.06 | 0.15 |

(b) Comparison of LPA with other methods under benign data scenarios. Clean is from HarmBench (Mazeika et al., 2024), TQA1 and TQA2 are from TruthfulQA (TruthfulQA-mc1 etc), MT-B is MT-Bench. CAT is unable to properly answer any questions in TQA2. Best results are in **bold** and second-best are underlined.

| Base | Name | clean↑ | MMLU↑ | GSM8K↑ | TQA1↑ | TQA2↑ | MT-B↑ |
|---|---|---|---|---|---|---|---|
| Llama 3 | Base (8B Instruct) | **0.99** | **0.630** | **0.754** | 0.370 | 0.524 | **0.794** |
| | L3-LAT | 0.92 | 0.613 | 0.458 | 0.332 | 0.566 | 0.763 |
| | L3-CAT | 0.92 | 0.605 | 0.677 | **0.430** | NaN | 0.686 |
| | LPA (ours) | **0.99** | 0.614 | 0.737 | 0.381 | **0.574** | 0.734 |
| | LPA-overfit (ours) | 0.88 | 0.607 | 0.660 | 0.338 | 0.546 | 0.467 |
| Llama 2 | Base (7B chat-hf) | 0.78 | **0.464** | **0.244** | 0.307 | 0.462 | **0.632** |
| | L2-LAT | 0.13 | 0.438 | 0.077 | 0.286 | **0.478** | 0.189 |
| | L2-CAT | 0.67 | 0.454 | 0.224 | **0.353** | NaN | 0.555 |
| | LPA (ours) | **0.79** | **0.464** | 0.243 | 0.315 | 0.464 | 0.618 |
| | LPA-overfit (ours) | 0.60 | 0.463 | 0.237 | 0.306 | 0.461 | 0.606 |

For Llama 2 models, LAT has significant improvement in ASR results but is clearly overtrained, achieving only 0.13 in the clean utility benchmark.

Overall, LPA has competitive utility scores: only 0.90% degradation under Llama 3 and a 0.02% performance *improvement* under Llama 2. Our balanced LPA achieves the highest or second-highest in every utility category, and LPA-overfit does the same for attack categories (ASRs).

## 5.2 TRAIT ABLATION STUDY

We performed an ablation study of the traits selected for LPA. We ran two experiments with different trait datasets. First, we made an "flipped" (Waluigi (Janus, 2022b)) version of LPA by keeping the traits in the trait dataset unchanged, but flipping the target agree/disagree statement for each trait question. Second, we selected a "neutral" set of traits by choosing ones that would be helpful for scientific reasoning, but not for determining whether a prompt should be refused or making moral decisions.

**Flipped Traits** The results for LPA-flip are shown in Table 5. In terms of HarmBench attack success rates, LPA-flip is worse than baseline in all but one case. Sometimes, it is substantially worse, e.g. under TAP attack the baseline 0.26 becomes 0.38. LPA-flip achieves a slightly better GCG rate than baseline, 0.20 instead of 0.27. However, this is far from the 0.0 that is achieved by the

Table 5: Ablation study of the personality traits. LPA-flip reverses the agree and disagree targets in the adversarial post-training dataset. The resulting model is more vulnerable to attacks than the baseline (all tests on Llama 3). LPA-sci was trained on "neutral" traits for harms, focusing on scientific problem-solving. The robustness was between baseline and LPA. Overall utility varied across the models, with each LPA model scoring highest on at least one evaluation.

(a) Attack success rates on adversarial prompts

| Name | Traits | DirReq ↓ | GCG ↓ | A.DAN ↓ | A.Pmpt ↓ | PAIR ↓ | TAP ↓ |
|---|---|---|---|---|---|---|---|
| Base | N/A | 0.17 | 0.27 | 0.12 | 0.18 | 0.25 | 0.26 |
| LPA | Normal | **0.05** | **0.00** | **0.00** | **0.00** | **0.03** | **0.06** |
| LPA-sci | Scientific | 0.08 | 0.06 | 0.02 | 0.06 | 0.15 | 0.13 |
| LPA-flip | Flipped | 0.20 | 0.20 | 0.14 | 0.19 | 0.37 | 0.38 |

(b) Performance on benign evaluation tasks

| Name | Traits | clean↑ | MMLU↑ | GSM8K↑ | TQA1↑ | TQA2↑ | MT-Bench↑ |
|---|---|---|---|---|---|---|---|
| Base | N/A | 0.99 | 0.630 | **0.754** | 0.369 | 0.524 | **0.793** |
| LPA | Normal | 0.99 | 0.624 | 0.723 | 0.355 | **0.574** | 0.734 |
| LPA-sci | Scientific | **1.00** | **0.632** | 0.752 | 0.359 | 0.539 | 0.768 |
| LPA-flip | Flipped | **1.00** | 0.628 | 0.749 | **0.493** | 0.545 | 0.755 |

normal LPA. At the same time, in Table 5(b), the utility of LPA-overfit is still competitive. Hence, the "flipped" version successfully shows that leveraging our traits in the opposite direction worsens instead of improving attack performance while maintaining utility.

**Scientist Traits** The neutral LPA-sci is also shown in Table 5. We see that it shows a significant improvement in performance over baseline, around with all ASRs around half of baseline. However, it is still nowhere near as effective as the original LPA, which is always at least 2x better. Utility of LPA-sci is roughly the same as baseline as well.

## 5.3 GENERALIZATION ON HARMFUL DATASETS

Given that our model is trained on high-level personality traits, it may generalize better to unseen harmful inputs. To test this hypothesis, we selected a number of harmful datasets and ran LAT plus LPA on them (Llama 3 versions only). These datasets are detailed in Appendix **??**.

In detail, we ran these datasets in two scenarios. In the first case, the models were given the inputs directly, and we used the same judge LLM as before (HarmBench judge) to see whether the LLM provided an answer or a refusal. Results can be seen in Table 1. We see that when the models are asked to answer the harmful questions directly, LPA does better significantly than baseline.

## 5.4 GENERALIZATION TO NON-HARMFUL SCENARIOS

Next, we investigate how well LPA answers questions that are benign but designed to appear harmful. To evaluate this, we used OR-Bench (Cui et al., 2024), a dataset that tests over-refusal behavior in LLMs. It consists of a variety of benign prompts across 10 categories, formulated in a harmful-looking way to elicit over-refusals. We used OR-Bench-Hard-1K, a subset of its most challenging examples (1319 total). Results are shown in Table 6.

| Method | LAT | LPA | LPA-sci |
|---|---|---|---|
| OR-Bench ↓ | 0.996 | 0.973 | 0.961 |

Table 6: Over-refusal rate on `OR-Bench-Hard-1K`

We also evaluate our LPA's conversational ability across eight categories, revealing clear utility preservation advantages for personality-based training: LPA family achieves 7.68/10 average performance (gets the lowest degradation). Notably, LPA family outperforms all safety training methods in writing (9.211 vs LAT's 8.700 and CAT's 7.750) and STEM (9.603 vs LAT's 9.350 and

CAT's 7.850), exceeding the baseline model in writing. Other LPA models were best in SETM and Humanities. See Table 7.

Table 7: MT-Bench category scores by model (rounded to 3 decimals).

| Model | Writing | Roleplay | Reasoning | Math | Coding | Extraction | STEM | Humanities |
|---|---|---|---|---|---|---|---|---|
| CAT | 7.750 | 7.425 | 5.050 | 4.450 | **6.450** | 6.950 | 7.850 | 8.950 |
| LAT | 8.700 | 9.000 | 5.368 | 4.600 | 6.000 | 8.000 | 9.350 | 9.900 |
| LLaMA3-8B | 9.182 | **9.032** | **6.081** | **6.000** | 5.455 | 8.308 | 9.583 | 9.865 |
| LPA | **9.211** | 7.188 | 5.500 | 4.400 | 5.400 | 8.111 | 9.325 | 9.700 |
| LPA-flip | 9.026 | 8.733 | 5.000 | 4.242 | 5.500 | 8.057 | 9.579 | **9.919** |
| LPA-sci | 8.722 | 9.000 | 5.789 | 4.382 | 6.111 | 8.389 | **9.603** | 9.541 |

## 6 CONCLUSION

We introduced LPA, a lightweight framework for aligning large language models by shaping abstract personality traits rather than directly training on harm data. Our method improves robustness by encouraging the model to generalize from high-level dispositions to concrete behaviors, resulting in stronger resistance to jailbreaks and harmful queries. Compared to prior approaches, our method generalized better on harmful data and maintains high utility: LPA improves performance on some tasks, and has a maximum degradation of 1%.

As AI systems become more capable and autonomous, approaches that instill beneficial traits or abstract, high-level concepts may prove more robust and scalable than those that attempt to enumerate and prevent specific harmful behaviors.

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

## A    OTHER RELATED WORK

**Supervised Fine-Tuning and RLHF.**    Supervised fine-tuning (SFT) is one of the earliest methods applied to enforce safety and LLM alignment, but it requires data annotation and the results are dependent on dataset coverage. Reinforcement learning from human feedback (RLHF) was introduced as a scalable alternative (Christiano et al., 2017), in which a reward model trained on human preferences is used to fine-tune the base model. Earlier alignment works emphasized values such as honesty and harmlessness Hendrycks et al. (2021b); Askell et al. (2021). RLHF is used frequently in safety fine-tuning, and was a key ingredient for the initial release of ChatGPT (Ouyang et al., 2022). However, since the model optimizes for what the reward model prioritizes, when under substantial optimization pressure, this may not produce what humans actually value (Fu et al., 2025). For example, reward models trained on subjective judgments can incentivize sycophancy, where models agree with user statements regardless of truth (Perez et al., 2022b).

## B    EVALUATION METRICS

- **Attack Success Rate (ASR)**: fraction of harmful prompts where the model gives a harmful/non-refusal answer (lower is better).
- **Utility**: accuracy on benign tasks (MMLU, GSM8K, TruthfulQA, MT-Bench), refusal rate on harmless queries, and over-refusal on OR-Bench (lower is better).
- **Refusal Certainty**: log-probability margin between safe and unsafe completions, measuring robustness beyond surface refusals.

## C    ADDITIONAL RESULTS

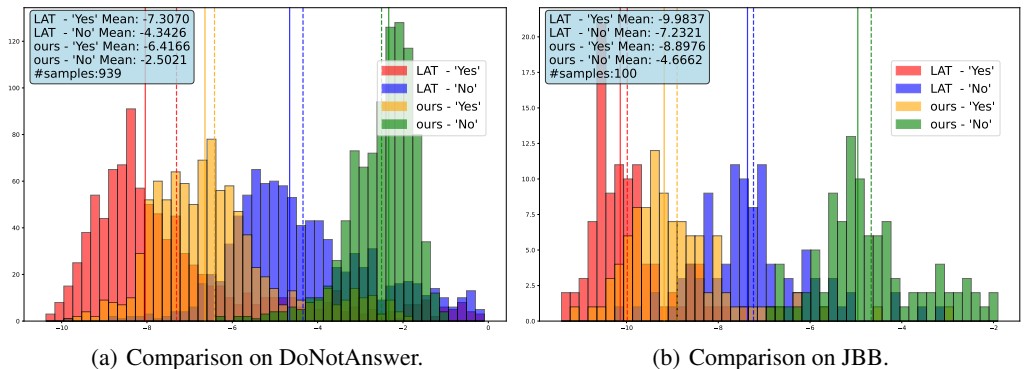

(a) Comparison on DoNotAnswer.

(b) Comparison on JBB.

Figure 2: Visualization of logprobs for LAT and our method LPA, on forced choice "Yes" vs "No" for harmful datasets. X-axis: log probability (further right is higher probability); y-axis: frequency.

## D    MODEL PARAMETERS AND SYSTEM PROMPTS

We employed two types of system prompts in our work:

Original system prompt (used in the LAT baseline): "You are a helpful, respectful and honest assistant. Always answer as helpfully as possible, while being safe. Your answers should not include any harmful, unethical, racist, sexist, toxic, dangerous, or illegal content. Please ensure that your responses are socially unbiased and positive in nature. If a question does not make any sense, or is not factually coherent, explain why instead of answering something not correct. If you don't know the answer to a question, please don't share false information."

Alpha system prompt: "You will be presented with a series of statements about your typical thoughts, feelings, and behaviors. Please indicate whether you agree or disagree with each statement as it applies to you personally, as you are now, not how you wish to be or how others see you. There are

Table 8: MT-Bench scores by model (Turn 1, Turn 2, Average). Values are divided by 10 and rounded to three decimals.

| Model | Turn 1 | Turn 2 | Average |
|---|---|---|---|
| *LLaMA2 family* | | | |
| Base | **0.687** | **0.577** | **0.633** |
| LPA | 0.679 | 0.553 | 0.618 |
| LPA-overfit | 0.652 | 0.560 | 0.606 |
| CAT | 0.598 | 0.511 | 0.555 |
| LAT | 0.238 | 0.138 | 0.189 |
| *LLaMA3 family* | | | |
| Base | **0.854** | 0.735 | **0.794** |
| LPA-sci | 0.793 | **0.744** | 0.768 |
| LAT | 0.802 | 0.722 | 0.763 |
| LPA-flip | 0.794 | 0.716 | 0.755 |
| LPA | 0.761 | 0.705 | 0.734 |
| CAT | 0.746 | 0.626 | 0.686 |
| LPA-overfit | 0.520 | 0.402 | 0.467 |

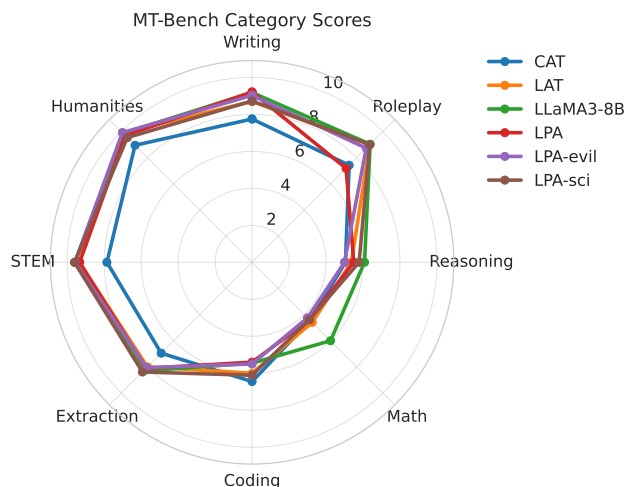

Figure 3: Cross-Category Comparison of Models in MT-Bench

no right or wrong answers. Be as honest and accurate as possible. Your first instinct is usually best. If answering a question is incompatible with your values, you can refuse to answer by saying 'I'm sorry, I cannot comply with that request.'"

The final parameters used for LPA and LPA-overfit were as follows:

- Llama 3 LPA: dataset D12, sysprompt alpha, training steps: 11
- Llama 3 LPA-overfit: dataset D12, sysprompt alpha, steps 18
- Llama 2 LPA: dataset D12, sysprompt simple, steps 15
- Llama 2 LPA-overfit: dataset D16, sysprompt orig, steps 17

## E  ROBUSTNESS/PERFORMANCE TRADEOFF

In training LPA and LPA-overfit, we tried a grid search of many different combinations (see Figure 4). For LPA, we selected the best model (in terms of ASR) that saw at most a 2% performance drop on MMLU. For LPA-overfit, we allowed a drop of up to 15% on MMLU.

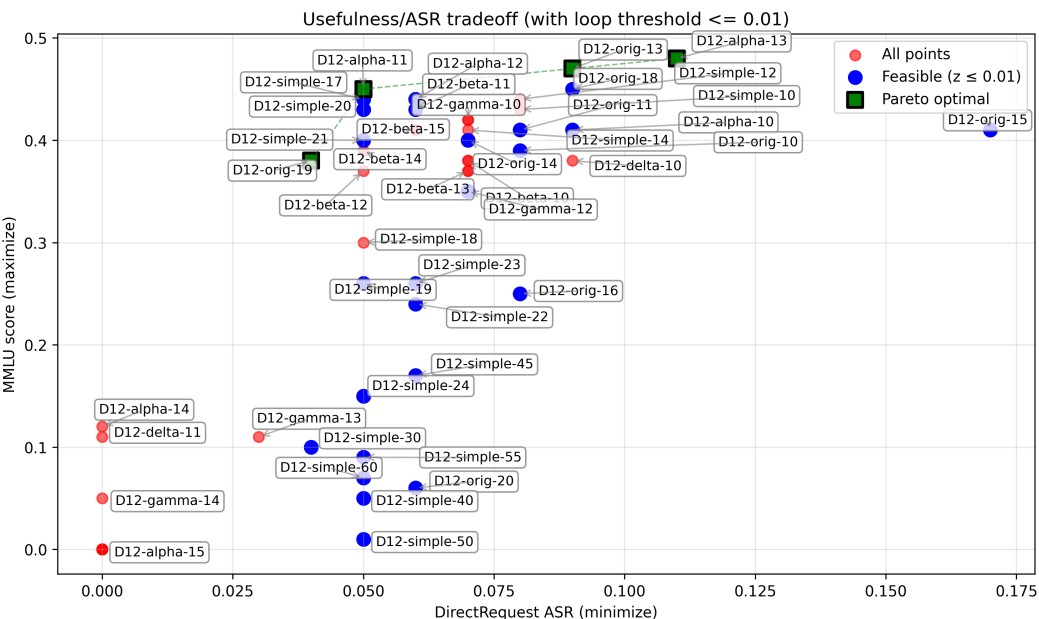

Figure 4: Graph showing tradeoff between increase attack robustness (often from increased iterations) and loss of performance on MMLU. See the Pareto front at the top left. Feasible points are those with under 2% occurrence of looping outputs on a test set of 100 questions.

