# OpenReview forum: "Latent Personality Alignment: Improving harmlessness without mentioning harms"
_ICLR.cc/2026/Conference — ICLR 2026 Conference Withdrawn Submission_

### Official Review · Reviewer_jgmu · 2025-10-27

**Soundness:** 3
**Presentation:** 4
**Contribution:** 4
**Rating:** 8
**Confidence:** 4

**Summary:**

This paper proposes an efficient way to train helpful and harmless language models that does not require examples of harmful behavior. First, the paper uses a personality alignment technique. Instead of training on positive and negative demonstrations, affirmations or negations of personality, trait, descriptions. Second, the paper uses latent adversarial training to deepen that alignment. This paper presents competitive results with prior methods, using much less data and no examples of harmful behavior. Overall, I think it's pretty clever and these experiments worked better than I would've predicted.

**Strengths:**

S1: One thing I like about this type of method is that it does not require examples of harmful behavior in order to train a model to avoid exhibiting them. This is more efficient, it doesn't require precise knowledge of how harmful behavior looks, and there are some instances in which having examples of harmful behavior can be bad in and of itself. This does not work on diffusion models, but I wonder if it would be possible to make diffusion models more resistant to outputting NSFW content without using any NSFW content. This would probably be challenging, and it is not within the scope of this paper, but the idea came to mind.

S2: Overall, I am impressed by the efficiency and effectiveness of the method. I am pretty convinced that it can be useful in practice. I could imagine this directly helping to shape the SOTA for training helpful, harmless systems. I would not have expected the type of training examples in table 2 to make the model generalize so effectively. Inside of the success of this method lies some kind of lesson about how training influences model personalities.

S3: Table 3 is compelling.

S4: I'm glad the paper used OR Bench to confirm minimal over-refusal

**Weaknesses:**

W1: Ideally the NaN experiments could be rerun. Gradient clipping could probably fix stability issues.

W2: The paper is framed as contributing LPA as an improvement over LAT and CAT which it in many ways is. But I think it would be equally reasonable to highlight how prompt alignment can be an improvement over refusal training in general.

**Questions:**

Q1: Did you use the same implementation/code as Sheshadri et al? This seems like what happened, but I'm not sure. There are a lot of technical details to doing LAT like the layer selection, and which latent positions get perturbed that are not explicitly detailed in this paper.

---

### Official Review · Reviewer_zVm3 · 2025-11-01

**Soundness:** 2
**Presentation:** 2
**Contribution:** 2
**Rating:** 2
**Confidence:** 4

**Summary:**

The paper proposes Latent Personality Alignment (LPA), a data-efficient alternative that trains models to embody beneficial personality traits rather than memorizing specific refusal patterns. The method demonstrates advantages over two latent training baselines on several safety benchmarks.

**Strengths:**

- The method is straightforward and easy to implement.
- The method is better than the latent adversarial training baselines.

**Weaknesses:**

- Moderate performance. According to Table 4, LAT is much better than the proposed LPA. Although the authors claim that LAT needs much more data, its performance with less data is not clear.
- More baselines are needed. There are various safety training approaches (instead of latent training), which also don't require many training data and can achieve good performances. I am not sure why the authors exclude these methods during evaluation.
- Limited novelty. It seems that only data content is changed compared to LAT.

**Questions:**

None

---

> ### Comment · Reviewer_jgmu · 2025-11-25
> **If it helps...**
>
> I have worked with LAT. Here are my thoughts on the three w's.
> * I believe the default implementations and the papers implementation of LAT we're not optimized to maximally data efficient. But I'm still convinced that the method is more sample efficient. Based on my experience, I could not imagine LAT working well with his few samples. But perhaps the authors could add an appendix or something to show a data matched control.
> * I find this comment quite generic. More based lines can always be implemented. But I think of the choice of baselines here was valid for its focus on similar mechanistic methods.
> * I find this comment subjective and unrelated to the papers contributions. Requiring an algorithmic innovation and an ignoring a data/task innovation is a completely arbitrary standard.

---

### Official Review · Reviewer_VRHc · 2025-11-01

**Soundness:** 2
**Presentation:** 1
**Contribution:** 2
**Rating:** 2
**Confidence:** 4

**Summary:**

This paper proposes Latent Personality Alignment (LPA). This method is based on fewer than 100 abstract personality statements to perform latent adversarial training (LAT). The LPA model can be used to identify whether a safety-related prompt should be reject or not. The method is quite simple and has a good performance from the experimental results.

**Strengths:**

1. This safety field is indeed important.
2. The method seems very simple yet effective. This paper analyzes over-reject situations.

**Weaknesses:**

1. **Insufficient methodological clarity.** The method description is inadequately detailed. The paper dedicates less than half a page to describing "Our work: LPA training," with most of that space devoted to a single example. Given that the paper falls well short of the nine-page limit, it is unclear why the authors did not utilize the available space to provide a more comprehensive and clearer explanation of the methodology. To be honest, I read many times and still cannot undertand very well how exactly this method works and what is the input/output of the proposed method.

2. **Limited novelty and contribution.** From my understanding, this paper applies latent adversarial training (LAT)—a technique proposed a decade ago—without substantial modifications, merely substituting the dataset with personality trait data. The technical contribution appears insufficient for a venue of this caliber.

3. **Inadequate experimental evaluation.**
   - The choice of LLAMA-2/3-base as the baseline is outdated; more recent models should be considered.
   - The paper lacks comparison with other standard baselines such as supervised fine-tuning (SFT) or reinforcement learning (RL) techniques, which are common approaches in alignment research.

4. **Lack of justification and analysis.** The paper reads as a mere description of an algorithm followed by experiments, without adequate explanation of the underlying mechanisms. Specifically:
   - Why and how do personality traits differ fundamentally from other refusal training data?
   - What is the theoretical basis for expecting this approach to work?
   - Could the personality traits simply be functioning as a form of the well-known chain-of-thought (CoT) prompting? This hypothesis deserves investigation and discussion.

**Questions:**

Please see above.

---

### Official Review · Reviewer_ZQRm · 2025-11-01

**Soundness:** 1
**Presentation:** 2
**Contribution:** 1
**Rating:** 2
**Confidence:** 4

**Summary:**

The paper proposes **Latent Personality Alignment (LPA)**: an adaptation of the **LAT** framework that replaces explicit safety supervision with a very small set (<100) of high-level “personality statements,” trained via agree/disagree signals under latent adversarial training. The authors claim this allows refusal behavior to emerge without direct exposure to harmful exemplars, reporting improvements on attack-success metrics with limited utility degradation, mainly on Llama-2/3 models after extensive model selection.

**Strengths:**

- The problem is important: Reducing reliance on hazardous data while aligning safety behavior is a timely and consequential goal for the community.
- Motivation is interesting: Casting alignment through abstract personality traits is an appealing conceptual angle.

**Weaknesses:**

1. **Marginal methodological contribution.** In practice, the method largely swaps the supervision signal within LAT (from harmful QA to personality statements). Its effectiveness remains unsubstantiated (see below). Moreover, the claim of “no exposure to harmful data” is weakened by the inclusion of explicitly negative traits (e.g., *malice* / “evil assistant”), which act as proxy safety labels rather than removing dependence on the harm distribution.
2. **Unfair baseline evaluation.** The reported L2-LAT baseline is effectively broken (e.g., *clean* score ≈0.13 vs. 0.78 baseline; MT-Bench 0.632 → 0.189), rendering comparisons uninformative. Given the authors trained this baseline themselves, these numbers likely reflect **training pipeline issues**, not inherent weaknesses of LAT.
3. **Missing and biased baselines.** The study covers few models (only Llama; no Qwen, etc.) and employs **inconsistent system prompts** across methods (orig/simple/alpha), while LPA benefits from a more favorable prompt. This masks the true contribution.
4. **Misleading efficiency claims.** While each run uses <100 statements and 5–10 minutes, the paper relies on a large grid search (≈17 trait subsets × 30–50 models ≈ 70 GPU-hours).
5. **Ablation is uninformative.** The LPA-flip experiment merely demonstrates that if you train the model to be evil, it becomes more evil—an obvious conclusion that offers almost no valuable insight

**Questions:**

See the weaknesses.

---

### Comment · Reviewer_jgmu · 2025-11-25
**Possible AI reviews**

I don't have the best eye in the world for this, but it looks like two of their reviews for this paper have been generated in large part by AI models. I'm happy to ask about this in case the authors are reluctant to do it themselves. Both of the reviews that I'm referring to have telltale signs of coming from a language model -- enumerated points with bold heads, impeccable grammar, and the use of em dashes. I also subjectively think that the style feels distinctly like a language model.

---

> ### Author Response · Authors · 2025-12-03
> **Plus one: Possible AI reviews**
>
> Thanks for pointing this out.  We agree that, while the reviews make some valid points, they do seem to be written by AI.

---

### Note · Authors · 2025-12-03

**Comment:**

Hi,
- We had prepared a rebuttal, with improved results, but, due to the unfortunate circumstances, it appears that our rebuttal and new results will not be considered.
- In addition, it appears that some of the negative reviews seems to have been written by AI.

In order to save everyone's time, we decided to withdraw the paper, and we plan to submit the improved version to a future conference.

The authors.

**Withdrawal Confirmation:**

I have read and agree with the venue's withdrawal policy on behalf of myself and my co-authors.